# Physico-Chemical and In Vitro Characterization of Chitosan-Based Microspheres Intended for Nasal Administration

**DOI:** 10.3390/pharmaceutics13050608

**Published:** 2021-04-22

**Authors:** Csilla Bartos, Patrícia Varga, Piroska Szabó-Révész, Rita Ambrus

**Affiliations:** Faculty of Pharmacy, Institute of Pharmaceutical Technology and Regulatory Affairs, University of Szeged, 6726 Szeged, Hungary; varga.patricia@szte.hu (P.V.); revesz@pharm.u-szeged.hu (P.S.-R.); ambrus.rita@szte.hu (R.A.)

**Keywords:** nasal administration, spray-drying, chitosan, microsphere, meloxicam

## Abstract

The absorption of non-steroidal anti-inflammatory drugs (NSAIDs) through the nasal epithelium offers an innovative opportunity in the field of pain therapy. Thanks to the bonding of chitosan to the nasal mucosa and its permeability-enhancing effect, it is an excellent choice to formulate microspheres for the increase of drug bioavailability. The aim of our work includes the preparation of spray-dried cross-linked and non-cross-linked chitosan-based drug delivery systems for intranasal application, the optimization of spray-drying process parameters (inlet air temperature, pump rate), and the composition of samples. Cross-linked products were prepared by using different amounts of sodium tripolyphosphate. On top of these, the micrometric properties, the structural characteristics, the in vitro drug release, and the in vitro permeability of the products were studied. Spray-drying resulted in micronized chitosan particles (2–4 μm) regardless of the process parameters. The meloxicam (MEL)-containing microspheres showed nearly spherical habit, while MEL was present in a molecularly dispersed state. The highest dissolved (>90%) and permeated (~45 µg/cm^2^) MEL amount was detected from the non-cross-linked sample. Our results indicate that spray-dried MEL-containing chitosan microparticles may be recommended for the development of a novel drug delivery system to decrease acute pain or enhance analgesia by intranasal application.

## 1. Introduction

Nasal drug delivery provides an opportunity not merely to treat local pathological conditions (e.g., allergic rhinitis, nasal congestion) but also to deliver active pharmaceutical ingredients (APIs) to the systemic circulation or directly through the blood–brain barrier to the central nervous system [1]. The nose respiratory region is crucial from the aspect of systemic drug absorption. Drugs administered intranasally bypass the first-pass hepatic metabolism, thus side effects are avoided, and the large surface and the high vascularization of the mucosa cause the rapid onset of action [2,3]. Since it is an easily accessible, non-invasive, and painless option for systemic therapies, it is well accepted by patients [4]. However, there are some limitations that need to be taken into account. Firstly, mucosa sensitivity cannot be neglected, thus, drugs and excipients intended for intranasal delivery must not be irritants and definitely cannot be toxic [5]. The mucociliary clearance is a key determinant concerning the APIs residence time. The mucus layer renews in every 15–20 min interval, thus to prolong the APIs’ contact time, the use of mucoadhesive polymers can be considered. The low permeability of the mucosa raises another problem that needs to be solved [6,7,8].

Nasal sprays, drops, gels, and ointments are extremely popular and widely used. Unfortunately, few nasal powders are accessible on the market, however, they have highly beneficial properties over the aforementioned formulations. Since nasal powders do not contain moisture and their physical stability is better concerning liquid and semi-solid formulations, they can be prepared without using preservatives [9,10]. Moreover, they are eliminated slowly from the nasal cavity, because the better adhesion allows a longer time period for the API absorption [11]. Particle size, morphology, or rheological features must be taken into consideration during the nasal powder formulation [12,13].

Chitosan is a semi-synthetic polymer that is obtained by chitin deacetylation, which is found mostly in crustaceans or mushroom cell walls [14]. It plays a key role in the biomedical field due to its advantageous properties. Chitosan and its derivatives as micro- or nanoparticles can be used for targeted or controlled delivery of antibiotics, antitumor drugs, proteins, or vaccines. They are highly suitable for tissue engineering and wound healing based on their stimulating effect on cell proliferation and tissue regeneration. In terms of nasal administration, chitosan’s biocompatibility—which is due to the non-toxicity of its degradation products to the human body—and mucoadhesive characteristics are preferred [15,16,17]. Due to the cationic nature of chitosan, an ionic bond can be formed by the interaction between the negatively charged substructures of the mucus layer and chitosan, enabling mucoadhesion [18,19]. The positive charge interacts with tight junction- associated proteins as well, causing the distance growth between epithelial cells and enhancing the permeation property of chitosan [20]. Chitosan-based drug delivery systems are widely used for achieving controlled drug release. It has been reported that, by using cross-linking agents, an increased stability could be accomplished [21]. Glutaraldehyde and formaldehyde were used mainly as cross-linkers, but for their toxic quality, sodium tripolyphosphate (TPP) may be a more conspicuous alternative [22]. It possesses a negative charge, thus ionic bond is developed between TPP and chitosan [23,24].

Non-steroidal anti-inflammatory drugs (NSAIDs) are essential in relieving acute pain or enhancing analgesia as adjuvants to opioids [25,26]. The intranasal application of NSAIDs may offer an opportunity to attain a rapid analgesic effect by their absorption through the nasal mucosa to the systemic circulation [27]. During the formulation of NSAIDs, it is inevitable to solve their solubility problems, which can result in dose reduction that leads to decreased side effects together with their bioavailability improvement [28]. Several technological methods are available for modifying the physico-chemical properties and increasing the dissolution rate of NSAIDs [29,30,31]. Spray-drying is a one-step production method which can be applied to change the dissolution properties of a drug and provides an opportunity to prepare microspheres that match nasal requirements. This technique allows the control of particle properties such as their shape and size in a rapid and reproducible way [32,33]. It looks promising to create nasal formulations by spray-drying for pain relief with adequate dissolution properties, however, there is not any available literature on this topic thus far.

In our work, meloxicam (MEL) was chosen as an NSAID. It is used in joint disease therapy and serves as a favorable option because of its side effect profile. An MEL-containing nasal formulation may provide an opportunity to ease the pain alone or to potentiate the effects of opioids. In our previous research works, MEL- and meloxicam potassium monohydrate-containing spray and gel forms were prepared and investigated. The goal of this study was to design MEL-containing mucoadhesive intranasal microparticles to increase the residence time and the bioavailability of drugs by enhancing their dissolution and permeation. Chitosan microspheres were produced by spray-drying process, setting the parameters in order to acquire an energy-saving and quick preparation method. The effect of a lower inlet air temperature (90 °C)—lesser known in the literature—was compared with higher air temperatures. Furthermore, we optimized the composition of the formulation intended for nasal application by preparing MEL-incorporated chitosan-based microparticles and adding different amounts of TPP as a cross-linking agent. Particle size, morphological, and rheological properties of the products ensured nasal deposition. The physico-chemical properties as well as the in vitro dissolution and diffusion were determined and evaluated.

## 2. Materials and Methods

### 2.1. Materials

MEL was from EGIS Ltd. (Budapest, Hungary). Low molecular weight chitosan (Mw = 3800–20,000 Da) was obtained from Sigma Aldrich (Sigma Aldrich Co. LLC, St. Louis, MO, USA). TPP was purchased from Alfa Aeasar Co. (Alfa Aeasar GmbH & Co. KG, Karlsruhe, Germany). Dimethyl sulfoxide was from VWR Chemicals BDH Prolabo, and acetic acid was from Molar Chemicals Ltd. (Budapest, Hungary).

### 2.2. Methods

#### 2.2.1. Preparation of Spray-Dried Products

Optimizing process parameters, 1% acetic acid chitosan solution was spray-dried using Büchi Mini Dryer B-191 (Büchi, Flawil, Switzerland) applying inlet air temperatures of 90, 120, and 150 °C and pump rates of 5, 10, and 15 mL/min. Aspirator capacity was 75% (Table 1). Afterwards, to optimize the composition of the formulation, the feeding emulsions were prepared of 50 mL 1% chitosan solution, 3.75 mL 4% MEL-dimethyl sulfoxide (DMSO) solution, and 0, 1, or 2 mL of 1% aqueous solution of TPP, applying the optimal parameters (Table 2). The physical mixtures (PMs) of chitosan, MEL, and TPP were produced as the control samples in the same mass ratio similarly to the spray-dried products. After spray-drying, the percentage yield was determined.

#### 2.2.2. Size Distribution by Laser Diffraction

The particle-size distribution of the spray-dried samples was measured by laser scattering (Malvern Mastersizer Sirocco 2000, Malvern Instruments Ltd., Worcestershire, UK). The measurements were carried out at 3 bar pressure and 75% frequency, and air was used as a dispersion medium. Approximately 1 g of product was tested in one measurement, and each measurement was performed 3 times. D0.1, D0.5, and D0.9 values were determined as the diameter of the particles below which 10, 50, and 90 volume percentages of the particles existed.

#### 2.2.3. Scanning Electron Microscopy (SEM)

The shape and the surface morphology of the spray-dried particles were visualized by SEM (Hitachi S4700, Hitachi Scientific Ltd., Tokyo, Japan). Under an argon atmosphere, the samples were sputter-coated with gold-palladium in a high-vacuum evaporator with a sputter coater, and they were examined at 10 kV and 10 μA. The air pressure was 1.3–13 MPa.

#### 2.2.4. Density Measurement

The bulked and the tapped densities of the formulations were measured using the Engelsmann Stampfvolumeter (Ludwigshafen, Germany) [34]. A 10 cm^3^ cylinder was filled with 1.5–2.0 cm^3^ powder to calculate bulk density. Then, it was tapped 1000 times. Compared to the volume before and after the taps, we calculated the tapped density of the samples. We calculated the flow characters of the samples from the bulk (ρ_b_) and the tapped (ρ_t_) density (Equation (1)):(1)$$\mathrm{Carr}\;\mathrm{index}=\frac{\rho_{t}-\rho_{b}}{\rho_{t}}\;\times \;100$$

#### 2.2.5. Structural Analyses

The thermal analysis was executed with a Mettler Toledo DSC 821e (Schwerzenbach, Germany) system with the STARe program V9.1 (Mettler Inc., Schwerzenbach, Switzerland). Approximately 2–5 mg of samples were heated from 25 °C to 300 °C, applying 10 °C·min^−1^ heating rate under a constant argon flow of 10 L·h^−1^. Physical mixtures of chitosan, MEL, and TPP in the same mass ratio as the spray-dried samples contained were mixed in a Turbula mixer (Turbula WAB, Systems Schatz, Switzerland) at 50 rpm for 10 min and were applied as control samples.

XRPD was performed to investigate the physical state of MEL in the samples with a Bruker D8 Advance diffractometer (Bruker AXS GmbH, Karlsruhe, Germany) with Cu K λI radiation (λ = 1.5406 Å). The samples were scanned at 40 kV and 40 mA with an angular range of 3° to 40° 2θ. Si was used to calibrate the instrument. DIFFRACTPLUS EVA software was used to perform the manipulations: Kα2-stripping, background removal, and smoothing.

#### 2.2.6. Fourier-Transformed Infrared Spectroscopy (FT-IR)

For the purpose of determining whether cross-linking and incorporation were successful, the FT-IR spectra of the samples was recorded on an AVATAR330 FT-IR spectrometer (Thermo Nicolet, Unicam Hungary Ltd., Budapest, Hungary) in the interval 400–4000 cm^−1^ at an optical resolution of 4 cm^−1^. Samples were ground and compressed into pastilles at 10 t with 0.15 g of KBr.

#### 2.2.7. Rheological Investigations

Rheological measurements were carried out at 32 °C with HAAKE RheoStress 1 Rheometer (HAAKE GmbH., Hamburg, Germany). Cone and plate geometry was used to study the rheological profile of the samples. The flow curve of the samples was determined by rotation tests controlled shear rate. The shear rate was increased from 0.1 to 100 1/s in controlled rate mode.

#### 2.2.8. In Vitro Dissolution

The European Pharmacopoeia (6th Edition) paddle method (USP dissolution apparatus, type II Pharma Test, Heinburg, Germany) was applied to appoint the dissolution of MEL. A total of 50 mL of phosphate buffer solution (pH 5.6 ± 0.1) at 30 ± 0.5 °C was used as a dissolution medium. Taking into account the drug content of the microparticles, samples containing 6 mg of MEL were dispersed. The rotation speed of the paddles was 100 rpm. At predetermined intervals, the amount of dissolved MEL was determined by spectrophotometry (UNICAM UV/Vis Spectrometer, Cambridge, UK) at 364 nm. The in vitro drug release data of products were evaluated kinetically using various mathematical models such as zero order, first order, Higuchi, Hixon–Crowell, and Korsmeyer-Peppas model [35].

#### 2.2.9. In Vitro Permeability

The in vitro permeability of MEL was studied on a modified horizontal diffusion model which simulated the nasal cavity circumstances (Figure 1). Samples containing 6 mg of MEL were added to the donor phase (9 mL), which was simulated with nasal electrolyte solution (SNES) of pH 6.0 ± 0.1 (represented the nasal cavity) [36]. Half the amount of the SNES was put into the donor chamber and, with its other half, the sample was washed in the donor phase. PB of pH 7.40, which corresponded with the pH of the blood, was used as the acceptor phase (9 mL). The two chambers were divided by a synthetic membrane (Whatman^®^ regenerated cellulose membrane filter with 0.45 μm pores) that was soaked in isopropyl myristate before the investigation. It modeled the lipophilic mucosa between the phases. The temperature of the phases was 30 °C (Thermo Haake C10-P5, Sigma, Aldrich Co. and the rotation rate of the stir-bars was set to 100 rpm. The amount of MEL diffused to the acceptor phase was determined spectrophotometrically at 364 nm in real time with an AvaLight DH-S-BAL spectrophotometer (AVANTES, Apeldoorn, The Netherlands). Each measurement was carried out in triplicate.

## 3. Results and Discussion

### 3.1. Particle Size Distribution

The analysis of the results measured by laser diffraction revealed the fact that, by changing the process parameters, the average particle size of spray-dried products was approximately between 2–4 μm. Since the inlet air temperature and the pump rate did not have any effect on the size distribution of chitosan microspheres (Table 3), we chose the mild 90 °C inlet air temperature (requiring the lowest heat energy) and the relatively quick 10 mL/min pump rate to produce cross-linked and MEL-containing particles. At 15 mL/min pump rate, there was not sufficient time for the atomized drops to dry, thus they stuck to the column wall. The usage of TPP as a cross-linking material did not have any impact on the sizes of drug-free chitosan particles, however, there was a noticeable increase in the size of MEL-containing particles, especially when the volume of TPP solution was boosted (Table 4). Based on the literature data, the produced product size is considered to be an appropriate one for nasal administration [37]. The yields of the samples were between 38–64% concerning the MEL-free products, and they reached the 29–48% range regarding the MEL-containing microspheres.

### 3.2. Morphology of the Samples

The SEM images provided an indication of the microspheres morphology. Products formulated by using different amounts of TPP solution (0, 1.0, 2.0 mL) were investigated. Drug-free particles (Samples 4, 10, 11) had a hollow structure. Nearly spherical microparticles were observed in case of MEL-containing samples (Samples 12, 13, 14). Drug-containing samples in the presence of 0 or 1.0 mL TPP solution revealed a depressed surface morphology with holes. Microspheres cross-linked with 2.0 mL TPP solution exhibited a smooth surface (Figure 2).

### 3.3. Powder Rheology Properties

The rheological properties of powders have a key role in their processability. Moreover, the deposition of particles in the nasal cavity is inversely proportional to the density. Hence, the density of microspheres has a key role in getting into the required nasal region. The lower density of particles could offer better flowability and an improved deposition. The bulked and the tapped densities and, furthermore, the Carr index values of formulations are shown in Table 5. In case of drug-containing products, the tapped density was around 0.15 g/cm^3^, which was lower compared to the drug-free samples, predicting drug deposition in the required nasal regions. The Carr index results were in the range of 17 and 29, indicating the flowability, a parameter that is also responsible for the deposition.

### 3.4. Structural Characterization by DSC and XRPD

DSC was applied to study the crystallinity and the melting of MEL in physical mixtures and in spray-dried products. Sharp endothermic peaks of MEL were observed in the physical mixtures (around 256 °C) that corresponded to the melting point of MEL, indicating that, in these cases, MEL was crystalline (Figure 3a). Chitosan is an amorphous additive. The endothermic peaks of crystalline MEL disappeared; only the characteristic curve of chitosan was recognized regarding the spray-dried products containing TPP, revealing the presence of MEL in a molecularly dispersed form. The non-cross-linked sample (Sample 12) presented a reduced MEL peak intensity referring to the presence of its crystalline fraction.

XRPD was employed to investigate the physical state of drug-containing spray-dried samples and PMs as controls. The XRPD diffractograms of PMs demonstrated the crystalline structure of MEL, as established during DSC measurements. Its characteristic peaks were detected at 13.22, 15.06, and 25.7° (2Θ). The diffractograms of the spray-dried samples reconfirmed the presence of drugs, especially in the molecularly dispersed form. A few of the peaks of MEL appeared with a reduced but growing intensity with the decrease of TPP content, suggesting the presence of crystalline MEL (Figure 3b). The highest amount of crystalline MEL form was found where no cross-linking agent was applied.

### 3.5. FTIR Investigations

The intermolecular interactions of the microspheres were characterized by FT-IR (Figure 4a). Seven characterization peaks were observed in HMW chitosan–TPP microspheres at 3363.41, 2881.27, 1646.15 to 1653.24, 1376.47 to 1587.93, 1058.24 to 1064.48, 1026.87 to 1028.81, and 886.58 to 894.85 cm^−1^. These peaks could be defined as O–H from H-bonded, C–H stretch form aldehyde, C=N and N–H from amine I and amide II, -CH_3_ symmetrical deformation, C–N from amine, C–O stretching, and C–H from alkene or aromatic bonds, respectively [38,39]. Increasing the amount of TPP the peaks at 3363.41 cm^−1^ became broad, indicating an enhancement in hydrogen bonding. The peak at 1646.15 to 1650.20 cm^−1^ became larger in the presence of TPP compared to chitosan alone thanks to the electrostatic interaction between the amino groups in chitosan and the phosphoric groups in TPP [40]. The TPP peak at 1127.29 cm^−1^ disappeared after chitosan and TPP cross-linking due to the intermolecular interactions of chitosan and TPP. MEL-containing microparticles showed the characteristic absorption bands at specific wavenumbers (Figure 4b) The intensity of characteristic peaks of MEL at 3290.13, 1550.56, and 1265.14 cm^−1^ decreased because of drug incorporation to the microsphere.

### 3.6. In Vitro Dissolution Study

Before the dissolution studies, microspheres were dispersed in phosphate buffer (pH = 5.6), and the viscosity of samples was detected. Samples displayed shear-thinning behavior thanks to the orientation of the polimer chains in the flow direction. The viscosity of samples increased with increasing TPP amount.

The in vitro dissolution test was carried out at pH of 5.6 in phosphate buffer simulating the nasal conditions. The dissolution of raw MEL and of MEL from cross-linked and non-cross-linked samples was studied. Unprocessed MEL was used as a control; only 4.5% of it dissolved in 60 min (Table 6). The spray-dried samples revealed fast initial release in the first 15 min, which was followed by a slower stage. The presence of drugs in a molecularly dispersed form resulted in the rapid dissolution of API from the microspheres. The dissolved amount of MEL was decreased by the growth of TPP concentration. The lowest dissolved amount of drug was perceptible in the presence of 2 mL TPP. The highest amount of MEL—more than 90% during 1 h—was dissolved from the non-cross-linked Sample 12. This phenomena could be explained with the formation of cross-links only as the result of reaction between the phosphate and the amino groups of chitosan in the case of Sample 12 [41]. Adding TPP, the enhancement in hydrogen bonding and the electrostatic interaction between the amino and the phosphoric groups of chitosan and TPP kept MEL inside the microparticles.

During the analyses of the kinetics of drug release, the data were evaluated by correlation coefficient (R^2^). R^2^ values were used as the criteria to choose the best model to describe drug release from the products (Table 7). Because of the low solubility, the dissolution of raw MEL was slow and fitted the zero order kinetics model. In the case of TPP-free spray-dried product (Sample 12), the strongest correlation was shown with the first order kinetics model (Equation (2)):(2)$$\frac{M_{t}}{M_{\infty }}=1-e^{-k*t}$$
where M_t_ is the cumulative amount of drug released at time “t”; M_∞_ is the initial amount of drug in the dosage form; k is the release rate constant, revealing that the dissolution rate was concentration dependent. Approaching the saturation concentration, the dissolution slowed down. Concerning Samples 13 and 14, the drug release fit the Korsmeyer-Peppas model (Equation (3)):(3)$$\frac{M_{t}}{M_{\infty }}={k*t}^{n}$$
where n is a constant, which characterizes the transport mechanism of diffusion, indicating that the drug release mechanism from these samples was diffusion controlled by gelling and the slow erosion of the chitosan [42].

### 3.7. In Vitro Diffusion Study

Modified diffusion horizontal cell model was used for diffusion investigations. Figure 5 demonstrates that the rapid dissolution of MEL from a molecularly dispersed state in the case of the spray-dried products resulted in a faster diffusion and a higher permeated drug concentration in the acceptor phase. The lowest drug amount permeated related to raw MEL. The highest diffused concentration was observed from the product which did not contain TPP in its composition (approximately 45 µg/cm^2^). In the presence of TPP, chitosan formed a well-structured complex due to the intermolecular interactions, resulting in decreased swelling capacity of the polymer matrix and less drug dissolution and diffusion.

## 4. Conclusions

The aim of our work was to prepare MEL-containing spray-dried chitosan microspheres for nasal administration. The effect of the process parameters (inlet air temperature and pump rate) on the particle size and the morphology of the microspheres was studied. As a novelty, a lower inlet air temperature (90 °C) was investigated than usual. With the chosen parameters, cross-linked and MEL-containing samples were formulated. The physicochemical (particles size, shape, crystalline- and chemical structure) and the rheological properties of the microspheres were characterized, and the dissolution rate and the diffusion through the artificial membrane of the drug-containing powders were investigated.

The inlet air temperature and the pump rate did not have an effect on particle size distribution and morphology, therefore, the parameters that required the least energy (90 °C) and resulted in fast drying (10 mL/min) were chosen. Hereinafter, applying these parameters, MEL-containing samples were prepared adding different amounts of TPP solutions (0, 1, or 2 mL). The size of spray-dried MEL containing microparticles increased compared to the drug-free particles, however, the average particle size was between 2.9–5.6 μm, and they had a spherical habit. The density of microspheres (around 0.15 g/cm^3^) predicted drug deposition in the respiratory region of the nose. In the spray-dried samples, MEL was primarily in a molecularly dispersed state, however, concerning the non-cross-linked samples, a small crystalline fraction of MEL was observed. The MEL incorporation to the chitosan microparticles was successful. According to the in vitro dissolution and the permeability studies, the amounts of dissolved and diffused MEL were decreased by raising the concentration of TPP. In the case of microspheres that were formulated without TPP, more than 90% of drugs were dissolved during 1 h, and the same product showed the highest permeated drug amount (≈45 µg/cm^2^). It can be explained by the formed cross-links, thus chitosan retained MEL from dissolution and diffusion. For all three samples, the initial rapid dissolution was followed by a decelerating drug release.

Because of mucoadhesive and permeability-enhancer features of chitosan and fast and continuous dissolution and diffusion of molecularly dispersed MEL, formulated microspheres prepared by spray-drying may be recommended for further optimization in order to develop nasal dosage forms. After the dose settings and the choice and the setting of the medical device suitable for nasal powder delivery are determined, the drug delivery system may be suggested for relieving acute pain or as an adjuvant of analgesia through the nasal mucosa.

## Figures and Tables

**Figure 1 pharmaceutics-13-00608-f001:**
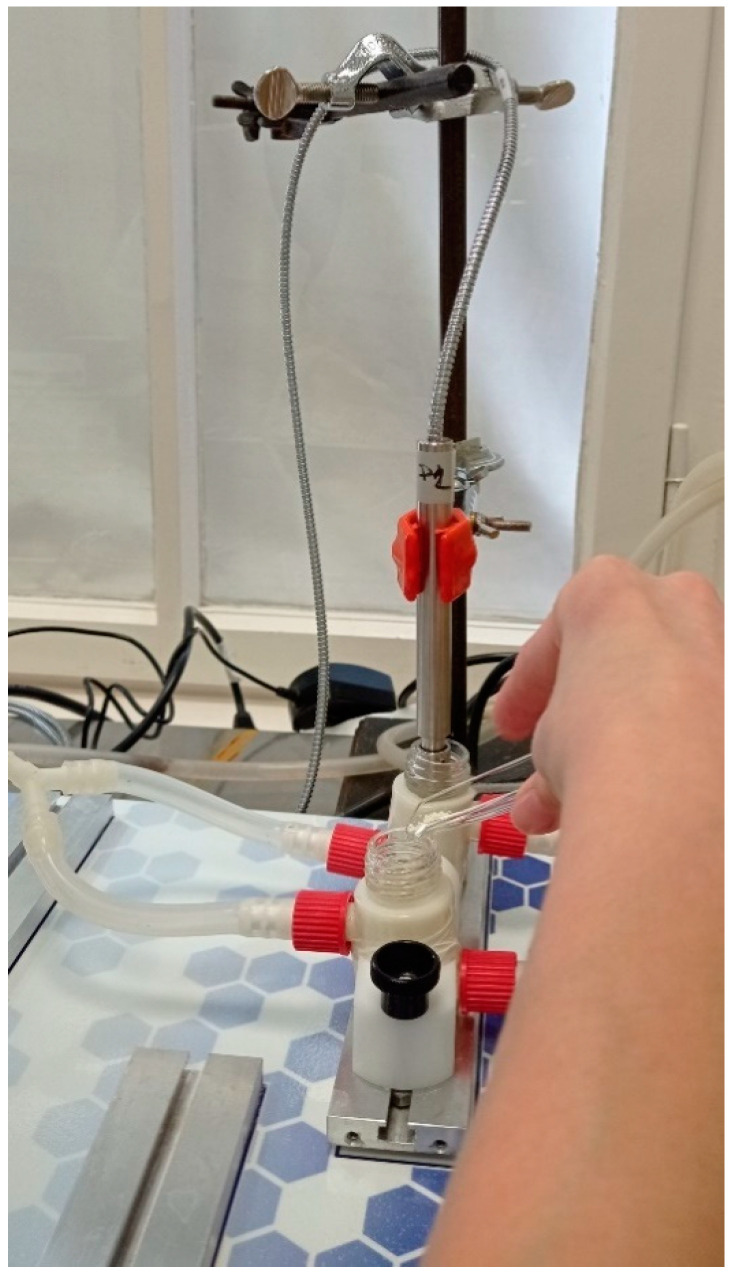
Illustration of in vitro permeability investigation.

**Figure 2 pharmaceutics-13-00608-f002:**
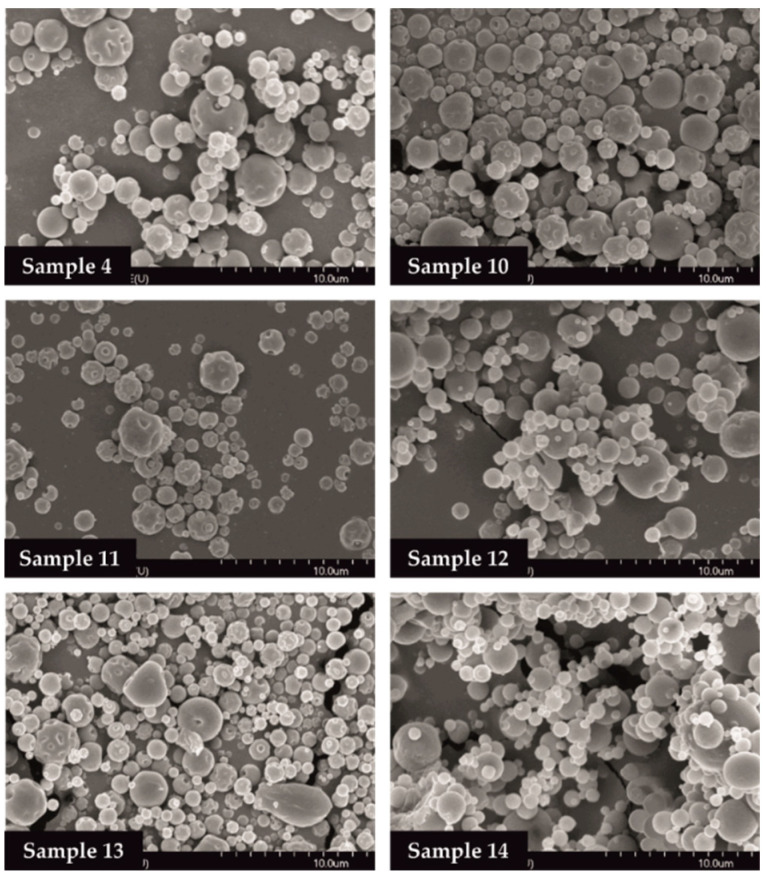
SEM images of spray-dried samples.

**Figure 3 pharmaceutics-13-00608-f003:**
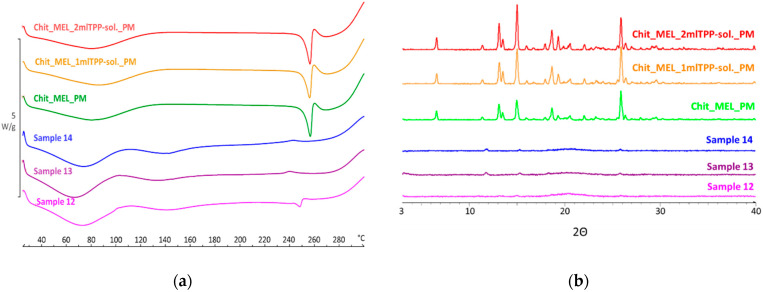
(**a**) DSC curves of PMs (Chit_MEL_2mlTPP-sol., Chit_MEL_1mlTPP-sol., Chit_MEL) and MEL-containing spray-dried samples; (**b**) XRPD patterns of PMs and MEL-containing spray-dried samples.

**Figure 4 pharmaceutics-13-00608-f004:**
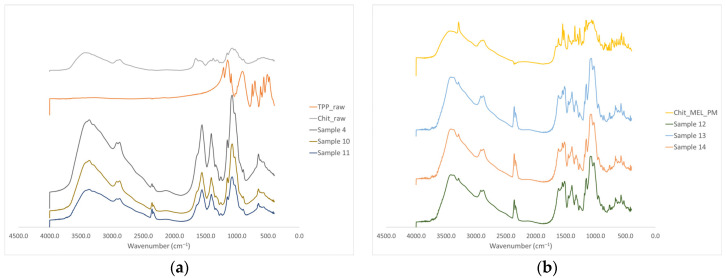
(**a**) FT-IR curves of the raw materials and the spray-dried samples without MEL, (**b**) FT-IR curves of the PMs and the MEL-containing spray-dried samples.

**Figure 5 pharmaceutics-13-00608-f005:**
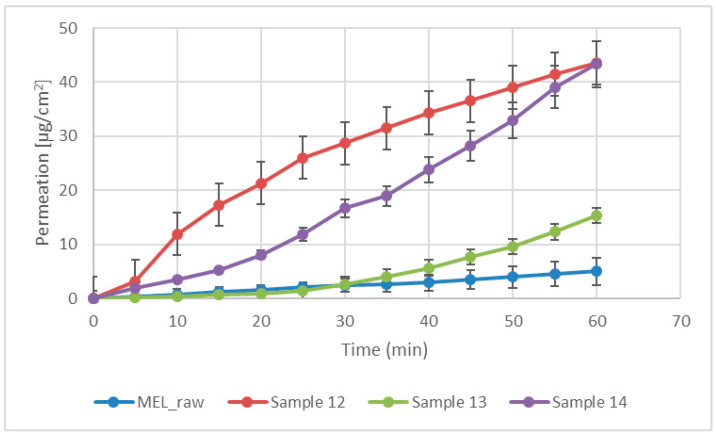
In vitro permeability of raw MEL and of MEL-containing spray-dried products.

**Table 1 pharmaceutics-13-00608-t001:** Spray-drying process parameters.

**Inlet Air Temperature (°C)**	90	120	150
**Pump Rate (mL/min)**	5	10	15

**Table 2 pharmaceutics-13-00608-t002:** Composition of solutions for spray-drying.

**1% Chitosan Solution (mL)**	50	50	50	50	50	50
**1% Aqueous TPP Solution (mL)**	-	1	2	-	1	2
**4% MEL-DMSO-Solution (mL)**	-	-	-	3.75	3.75	3.75

**Table 3 pharmaceutics-13-00608-t003:** Optimization of the process parameters.

Sample	1	2	3	4	5	6	7	8	9
Inlet air temperature (°C)	90	120	150	90	120	150	90	120	150
Pump rate (mL/min)	5	5	5	10	10	10	15	15	15
Aspirator (%)	75	75	75	75	75	75	75	75	75
D0.1 (μm)	1.044	1.446	1.529	1.176	1.255	1.241	1.115	1.274	1.369
D0.5 (μm)	2.374	3.669	3.736	2.466	2.815	2.701	2.263	2.629	2.889
D0.9 (μm)	5.216	8.535	9.032	5.102	5.903	5.519	4.744	5.195	5.664

**Table 4 pharmaceutics-13-00608-t004:** Optimization of the composition.

Sample	4	10	11	12	13	14
Inlet air temperature (°C)	90	90	90	90	90	90
Pump rate (mL/min)	10	10	10	10	10	10
Aspirator (%)	75	75	75	75	75	75
1% aqueous TPP-solution (mL)	-	1	2	-	1	2
MEL-DMSO-solution (mL)	-	-	-	3.75	3.75	3.75
D0.1 (μm)	1.176	1.243	1.103	1.269	1.426	1.617
D0.5 (μm)	2.466	2.595	2.419	2.965	3.757	5.575
D0.9 (μm)	5.102	5.234	5.138	7.211	9.461	15.995

**Table 5 pharmaceutics-13-00608-t005:** Powder rheology properties of the products.

Sample		4	10	11	12	13	14
Density (g/cm^3^)	Bulk	0.2490	0.1384	0.3112	0.1256	0.1176	0.1193
	Tap	0.6225	0.2214	0.5187	0.1507	0.1470	0.1670
Carr index (%)		60	38	40	17	20	29

**Table 6 pharmaceutics-13-00608-t006:** The percentage of dissolved drugs from raw MEL and MEL-containing spray-dried products.

Time (min)	Dissolved Drug (%)			
	Raw MEL	Sample 12	Sample 13	Sample 14
5	0.157 ± 0.01	50.15 ± 2.44	35.53 ± 2.51	48.85 ± 1.78
10	0.583 ± 0.03	60.24 ± 3.05	47.62 ± 3.01	60.96 ± 2.38
15	1.003 ± 0.05	68.18 ± 3.76	55.97 ± 3.41	75.20 ± 2.80
30	2.618 ± 0.13	78.37 ± 3.10	67.53 ± 3.92	79.93 ± 3.38
60	4.548 ± 0.23	82.38 ± 4.68	73.28 ± 4.1	93.65 ± 3.66

**Table 7 pharmaceutics-13-00608-t007:** R^2^ values of kinetic analysis of in vitro drug release using different models.

Model	Raw MEL	Sample 12	Sample 13	Sample 14
	R^2^	R^2^	R^2^	R^2^
Zero order	0.9927	0.7658	0.7309	0.7787
First order	0.989	0.9685	0.8626	0.8732
Higuchi	0.8021	0.7799	0.6757	0.8234
Hixon–Crowell	0.9892	0.8304	0.6634	0.7354
Korsmeyer-Peppas	0.9756	0.945	0.9528	0.9496

## Data Availability

Not applicable.

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
