# Peer review of "Physico-Chemical and In Vitro Characterization of Chitosan-Based Microspheres Intended for Nasal Administration"

_pharmaceutics, 2021, doi:10.3390/pharmaceutics13050608_

Round 1
Reviewer 1 Report
This manuscript presents the obtaining of chitosan microspheres by spray-drying process setting the parameters in order to acquire an energy-saving and a quick preparation method. The effect of a lower inlet air temperature (90 °C) – lesser-known in the literature – was compared with higher air temperatures. Furthermore, the authors optimized the composition of the formulation intended for nasal application by preparing meloxicam-incorporated chitosan-based microparticles and adding different amounts of sodium tripolyphosphate as a cross-linking agent. Particle size, morphological and rheological properties of the products ensured nasal deposition. The physico-chemical properties, in vitro dissolution and diffusion were determined and evaluated.
The experiments in this study are well planned and of good technical quality. However, several issues need to be addressed for the manuscript to be published in Pharmaceutics. To improve the manuscript I suggest the following comments:
- The authors should add several benefits of chitosan that make it suitable for biomedical applications. There are many articles in the literature that can be mentioned, such as:
Chitin and chitosan in selected biomedical applications, doi: 10.1016 / j.progpolymsci.2014.02.008
Double crosslinked interpenetrated network in nanoparticle form for drug targeting - preparation, characterization and biodistribution studies, doi: 10.1016 / j.ijpharm.2012.06.029
Biomedical Applications of Chitosan and Its Derivative Nanoparticles, doi: 10.3390 / polym10040462 - the equations must be rewritten, they are not legible
- the conclusions section should be written at the end of the discussions
Author Response
Reply to Referee comments
For Reviewer 1
Thank you very much for the valuable remarks. We greatly appreciate your advices. Here you can see listed all of the modifications made in the paper according to your suggestions (shown in blue colour in the text). English language of manuscript was checked.
- The authors should add several benefits of chitosan that make it suitable for biomedical applications. There are many articles in the literature that can be mentioned (…):
Thank you for your suggestion, the text is modified.
„Chitosan is a semi-synthetic polymer that is obtained by chitin deacetylation, which is found mostly in crustaceans or mushroom cell walls [14]. It plays a key role in the biomedical field due to its advantegous properties. Chitosan and its derivatives as micro- or nanoparticles can be used for the targeted or controlled delivery of antibiotics, anti-tumor drugs, proteins or vaccines. They are highly suitable for tissue engineering and wound healing based on their stimulating effect on cell proliferation and tissue regenera-tion. In terms of nasal administration, chitosan’s biocompatibility – which is due to the non-toxicity of its degradation products to the human body – and mucoadhesive charac-teristics are preferred [15-17].”
- Sinha, V.R.; Singla, A.K.; Wadhawan, S.; Kaushik, R.; Kumria, R.; Bansal, K.; Dhawan, S. Chitosan Microspheres as a Potential Carrier for Drugs. International Journal of Pharmaceutics 2004, 274, 1–33
- Zhao D., Yu S., Sun B., Gao S., Guo S., Zhao K. Biomedical Applications of Chitosan and ItsDerivative Nanoparticles. Polymers 2018, 10, 462.
- Anitha A., S.Sowmya S., Sudheesh Kumar P.T., Deepthi S., Chennazhi K.P., Ehrlich H., Tsurkan M., Jayakumar R. Chitin and chitosan in selected biomedical application. Prog. Polym. Sci. 39 (2014) 1644–1667.
- J ̆atariu(Cadinoiu) A.N., Holban M.N., Peptu C.A., Sava A., Costuleanu M., Popa M. Double crosslinked interpenetrated network in nanoparticle form for drug targeting—Preparation, characterization and biodistribution studies. Int. J. Pharm. 436 (2012) 66–74.
- The equations must be rewritten, they are not legible.
Thank you for your comment, the equations are rewritten.
|
(1) |
|
(2) |
|
|
|
|
|
(3) |
- The conclusions section should be written at the end of the discussions.
Thank you for your comment, the structure of the manuscript is redesigned.
„3. Results and discussion
[…]
- Conclusions
The aim of our work was to prepare MEL-containing spray-dried chitosan micro-spheres for nasal administration. The effect of the process parameters (inlet air tempera-ture and pump rate) on the particle size and morphology of the microspheres was studied. As a novelty, a lower inlet air temperature (90 °C) was investigated than usaually. With the chosen parameters cross-linked and MEL-containing samples were formulated. The physicochemical (particles size, shape, crystalline- and chemical structure) and rheologi-cal properties of the microspheres were characterized, and the dissolution rate and diffu-sion through the artificial membrane of the drug-containing powders were investigated.
The inlet air temperature and pump rate did not have an effect on the particle size distribution and morphology, therefore the parameters that required the least energy (90 °C) and resulted in fast drying (10 ml/min) were chosen. Hereinafter, applying these pa-rameters, MEL-containing samples were prepared adding different amounts of TPP solu-tions (0, 1 or 2 ml). The size of spray-dried MEL containing microparticles increased compared to the drug-free particles, however, the average particle size was between 2.9-5.6 μm, and they had a spherical habit. The density of microspheres (around 0.15 g/cm3) pre-dicted drug deposition in the respiratory region of nose. In the spray-dried samples, MEL was primarily in a molecularly dispersed state, however, concerning the non-cross-linked samples a small crystalline fraction of MEL was observed. The MEL incorporation to the chitosan microparticles was successful. According to the in vitro dissolution and permea-bility studies, the amount of dissolved and diffused MEL was decreased by raising the concentration of TPP. In case of microspheres that were formulated without TPP more than 90% of drug was dissolved during 1 h, and the same product showed highest per-meated drug amount (≈45 ug/cm2). It can be explained by the formed cross-links so that chitosan retained MEL from dissolution and diffusion. For all three samples, the initial rapid dissolution was followed by a decelerating drug release.
Because of the mucoadhesive and permeability-enhancer features of chitosan and the fast and continuous dissolution and diffusion of molecularly dispersed MEL, formulated microspheres prepared by spray-drying may be recommended for further optimization in order to develop nasal dosage form. After the dose settings and the choice and setting of medical device which is suitable for nasal powder delivery, the drug delivery system may be suggested for relieving acute pain or as adjuvant of analgesia through the nasal mucosa.”
- 04. 2021. Szeged, Hungary Csilla Balla-Bartos PhD
Assistant Professor

Reviewer 2 Report
This manuscript is not suitable for publication because there is many problems should be improved. This manuscript needs revision described below. If Pharmaceutics accept small novelty, this manuscript may be published after revision.
- I cannot understand the mean of “Micrometric”in the title. I think “Micrometric and” should be removed.
- Why does this in vitro permeability evaluation simulate nasal absorption? References for this method should be attached.
- There is no illustration for abbreviations “D0.1 and D0.5 and D0.9”.
- Texts in graphs are too small. We cannot read them.
- The horizontal axis (cm-1) for IR spectra are inverse. They should be reversed.
- There is no conclusion section in this manuscript.
Author Response
Reply to Referee comments
For Reviewer 2
Thank you very much for the valuable remarks. We greatly appreciate your advices. Here you can see listed all of the modifications made in the paper according to your suggestions (shown in red colour in the text, and blue colour for Reviewer 1). English language of manuscript was checked.
List of corrections:
- I cannot understand the mean of “Micrometric”in the title. I think “Micrometric and” should be removed.
Thank you for your suggestion, the title of the article is corrected to „Physico-chemical and in vitro characterization of chitosan-based microspheres intended for nasal administration”
- Why does this in vitro permeability evaluation simulate nasal absorption? References for this method should be attached.
Thank you for your suggestion. In the text was phosphate buffer solution (pH 5.6) which is corrected for simulated nasal electrolyte solution (SNES).
„Samples containing 6 mg of MEL were added to the donor phase (9 ml) which was simulated nasal electrolyte solution (SNES) of pH 6.0 ± 0.1 (represented the nasal cavity). [36]. Half amount of the SNES was put into the donor chamber and with its other half the sample was washed in the donor phase.”
During the in vitro permeability study simulated nasal electrolyte solution (SNES) was used as a donor phase. The SNES consisted of 8.77 g of NaCl, 2.98 g of KCl, and 0.59 g of anhydrous CaCl2 in 1000 mL of deionized water, the pH was set at 6.0 ± 0.1 with HCl (Jug et al., 2016). The question of the applicability of this method to nasal formulations has already been investigated in a study, in which the horizontal membrane diffusion model was compared to the vertical Franz diffusion cell model based on in vitro and ex vivo studies of permeability of meloxicam. It was found, that the horizontal membrane diffusion model is suitable to investigate nasal formulations (Horváth et al., 2015).
- Jug, M., Hafner, A., Lovrić, J., Kregar, M.L., Pepić, I., Vanić, Ž., Cetina-Čižmek, B., Filipović-Grčić, J. An overview of in vitro dissolution/release methods for novel mucosal drug delivery systems. Journal of Pharmaceutical and Biomedical Analysis 2018, 147, 350–366, doi:10.1016/j.jpba.2017.06.072.
Horváth, T., Ambrus, R., Szabó-Révész, P. Investigation of permeability of intranasal formulations using Side-Bi-Side horizontal diffusion cell. Acta Pharm. Hung. 85 (2015) 19-28.
- There is no illustration for abbreviations “D0.1 and D0.5 and D0.9”.
Thank you for your remark, the text is clarified.
„D0.1, D0.5 and D0.9 values were determined as the diameter of the particles below which 10, 50 and 90 volume percentage of the particles exist.”
- Texts in graphs are too small. We cannot read them.
Thank you for your remark, the graphs are modified.
|
|||||||||||
|
(a) |
(b) |
Figure 3. (a) DSC curves of PMs (Chit_MEL_2mlTPP-sol., Chit_MEL_1mlTPP-sol., Chit_MEL) and MEL-containing spray-dried samples; (b) XRPD patterns of PMs and MEL-containing spray-dried samples
|
(a) |
(b) |
Figure 4. (a) FTIR curves of the raw materials and spray dried samples without MEL, (b) FTIR curves of the PMs and MEL-containing spray dried samples
- The horizontal axis (cm-1) for IR spectra are inverse. They should be reversed.
Thank you for your remark, Figure 4. is corrected (you can find it in the answer to Question 4)
- There is no conclusion section in this manuscript.
Thank you for your comment, the structure of the manuscript is redesigned.
„ 3. Results and discussion
[…]
- Conclusions
The aim of our work was to prepare MEL-containing spray-dried chitosan micro-spheres for nasal administration. The effect of the process parameters (inlet air tempera-ture and pump rate) on the particle size and morphology of the microspheres was studied. As a novelty, a lower inlet air temperature (90 °C) was investigated than usaually. With the chosen parameters cross-linked and MEL-containing samples were formulated. The physicochemical (particles size, shape, crystalline- and chemical structure) and rheologi-cal properties of the microspheres were characterized, and the dissolution rate and diffu-sion through the artificial membrane of the drug-containing powders were investigated.
The inlet air temperature and pump rate did not have an effect on the particle size distribution and morphology, therefore the parameters that required the least energy (90 °C) and resulted in fast drying (10 ml/min) were chosen. Hereinafter, applying these pa-rameters, MEL-containing samples were prepared adding different amounts of TPP solu-tions (0, 1 or 2 ml). The size of spray-dried MEL containing microparticles increased compared to the drug-free particles, however, the average particle size was between 2.9-5.6 μm, and they had a spherical habit. The density of microspheres (around 0.15 g/cm3) pre-dicted drug deposition in the respiratory region of nose. In the spray-dried samples, MEL was primarily in a molecularly dispersed state, however, concerning the non-cross-linked samples a small crystalline fraction of MEL was observed. The MEL incorporation to the chitosan microparticles was successful. According to the in vitro dissolution and permea-bility studies, the amount of dissolved and diffused MEL was decreased by raising the concentration of TPP. In case of microspheres that were formulated without TPP more than 90% of drug was dissolved during 1 h, and the same product showed highest per-meated drug amount (≈45 ug/cm2). It can be explained by the formed cross-links so that chitosan retained MEL from dissolution and diffusion. For all three samples, the initial rapid dissolution was followed by a decelerating drug release.
Because of the mucoadhesive and permeability-enhancer features of chitosan and the fast and continuous dissolution and diffusion of molecularly dispersed MEL, formulated microspheres prepared by spray-drying may be recommended for further optimization in order to develop nasal dosage form. After the dose settings and the choice and setting of medical device which is suitable for nasal powder delivery, the drug delivery system may be suggested for relieving acute pain or as adjuvant of analgesia through the nasal mucosa.”
- 04. 2021. Szeged, Hungary Csilla Balla-Bartos PhD
Assistant Professor

Round 2
Reviewer 1 Report
The manuscript "Physico-chemical and in vitro characterization of chi-tosan-based microspheres intended for nasal administration", in its revised form, can be accepted for publication in Pharmaceutics. The authors have read the reviewers' recommendations carefully and considerably improved the manuscript.
Reviewer 2 Report
My points have been improved.